Prediction of influenza A virus-human protein-protein interactions using XGBoost with continuous and discontinuous amino acids information

Li Binghua 1 2 3
Li Xin 1 2 3
Li Xiaoyu 1 2 3
Wang Li 1 2 3
Lu Jun 4
Wang Jia 1 2 3 wang.jia@mail.hzau.edu.cn
1 College of Informatics, Huazhong Agricultural University , Wuhan , China
2 Hubei Key Laboratory of Agricultural Bioinformatics, Huazhong Agricultural University , Wuhan , China
3 Key Laboratory of Smart Farming for Agricultural Animals, Huazhong Agriultrual University , Wuhan , China
4 College of Engineering, Huazhong Agricultural University , Wuhan , China
Bolshoy Alexander
Electronic publication date: 2025 Jan 30
Publication date: 2025
Volume: 13
Electronic Location ID: e18863
Received 2024 Sep 25; Accepted 2024 Dec 23
Copyright: © 2025 Li et al.
Copyright year: 2025
Copyright holder: Li et al.
License: This is an open access article distributed under the terms of the Creative Commons Attribution License, which permits unrestricted use, distribution, reproduction and adaptation in any medium and for any purpose provided that it is properly attributed. For attribution, the original author(s), title, publication source (PeerJ) and either DOI or URL of the article must be cited.
License URL: https://creativecommons.org/licenses/by/4.0/

Keywords: Pathogen-host interaction (PHI), Protein-protein interaction (PPI), Influenza A virus, XGBoost, Machine learning, GO and KEGG

Funding: Central Universities of China 2662018JC034 The Fundamental Research Funds for the Central Universities of China (2662018JC034). The funders had no role in study design, data collection and analysis, decision to publish, or preparation of the manuscript.

==============================
Influenza A virus (IAV) has the characteristics of high infectivity and high pathogenicity, which makes IAV infection a serious public health threat. Identifying protein-protein interactions (PPIs) between IAV and human proteins is beneficial for understanding the mechanism of viral infection and designing antiviral drugs. In this article, we developed a sequence-based machine learning method for predicting PPI. First, we applied a new negative sample construction method to establish a high-quality IAV-human PPI dataset. Then we used conjoint triad (CT) and Moran autocorrelation (Moran) to encode biologically relevant features. The joint consideration utilizing the complementary information between contiguous and discontinuous amino acids provides a more comprehensive description of PPI information. After comparing different machine learning models, the eXtreme Gradient Boosting (XGBoost) model was determined as the final model for the prediction. The model achieved an accuracy of 96.89%, precision of 98.79%, recall of 94.85%, F1-score of 96.78%. Finally, we successfully identified 3,269 potential target proteins. Gene ontology (GO) and pathway analysis showed that these genes were highly associated with IAV infection. The analysis of the PPI network further revealed that the predicted proteins were classified as core proteins within the human protein interaction network. This study may encourage the identification of potential targets for the discovery of more effective anti-influenza drugs. The source codes and datasets are available at https://github.com/HVPPIlab/IVA-Human-PPI/.

Introduction

The influenza virus belongs to the genus Influenza within the family Orthomyxoviridae and is a type of pathogen that causes influenza (Ahmad et al., 2015). Due to its susceptibility to genetic recombination leading to antigenic variation, it has caused several pandemics and posed a serious threat to human public health (Shaw, 2011). In addition to pandemic influenza outbreaks, seasonal influenza epidemics occur annually in all regions of the world, resulting in approximately three to five million severe cases and 250,000 to 500,000 deaths annually (Huang et al., 2011). The influenza viruses are classified into four types: A, B, C and D. Among them, type A (IAV) has the greatest impact and the widest range of hosts (Lee, Wang & Park, 2018), being able to infect a wide range of mammals, including humans. Meanwhile, IAV is responsible for seasonal flu and all previous flu pandemics (Nogales et al., 2021). Since a virus depends heavily on the host cellular machinery for its replication, intermolecular interactions between the virus and host must be established for their propagation (Ain et al., 2020). By gaining insight into virus-host protein-protein interactions, scientists can identify important host factors enriched in the infection pathway that may be used as potential drug targets and develop antiviral drugs against these targets (Marques et al., 2019). Hence, the investigation of the interaction between proteins of the influenza A virus and human proteins holds significant importance in the diagnosis of influenza infections and the containment of influenza virus pandemics.

There are many traditional experimental methods for predicting PPIs, with yeast two-hybrid (Y2H) and tandem affinity purification-mass spectrometry (TAP-MS) being the two most prominent experimental approaches (Farooq et al., 2021). Although these traditional experimental methods have helped people to detect many unknown protein interactions, biological experiments can be affected by a number of factors that can lead to false positives in the experimental results (Deng et al., 2020), and they have the problems of high cost and large consumption of manpower and material resources, so the available and reliable virus-human PPI data are still insufficient (Yang et al., 2021). Therefore, validating traditional experimental methods through computational approaches and supplementing PPI data with prediction methods is imperative. The commonly employed computational methods are protein structure-motif interactions-based methods (DMI) (Evans et al., 2009), protein structure similarity-based methods (Tiwari et al., 2021), and machine learning (ML)-based methods (Zhang et al., 2017). Despite the gradual improvement of biological databases, the coverage of structural information for proteins is still incomplete. This makes the approach of using machine learning to study PPIs only based on protein sequence information stand out, as protein sequences are easily accessible. The human-virus PPI prediction task can be treated as a binary classification problem on the positive and negative samples. However, since there are no experimentally established non-interacting protein pairs (negative samples), careful selection of negative samples becomes crucial (Murakami & Mizuguchi, 2022). Another crucial challenge of ML-based methods is how to encode variable-length protein sequences into fixed-length numerical feature vectors that can be inputted into models to predict (You, Chan & Hu, 2015). Furthermore, the extraction of appropriate features from protein sequences is of paramount importance for accurately predicting PPIs (You et al., 2013). An effective feature representation should capture essential biochemical, structural, and sequence-level information from protein sequences to ensure that the model learns key interaction patterns. The more comprehensively the representation characterizes protein interactions, the more insights the model can extract, ultimately improving prediction accuracy.

Many researchers have conducted the construction of sequence-based predictive PPI models. Yang et al. (2020) applied doc2vec on the dataset of known PPIs between all viruses and human to represent protein sequences as low-dimensional but information-rich feature vectors, which allowed it to capture more contextual information from protein sequences, and obtained excellent prediction accuracy using Random Forest (RF) as classifiers. Liu et al. (2021) developed a prediction model for PPIs between hepatitis C virus (HCV) and humans by integrating features generated from the pseudo-amino acid composition. Dey & Mukhopadhyay (2019) used amino acid composition and conjoint triad to extract features from protein sequences and used support vector machine (SVM) model to predict PPIs between dengue and its human hosts.

The proposal of these methods proves the feasibility of using a sequence-based ML model for PPI prediction and shows the promise of providing a practical reference for further in-depth research on viruses in the future. However, current methods still have deficiencies. First, few studies have explored how to construct high-quality PPI datasets for training models, as there is no gold standard for positive and negative samples. Positive samples are often derived from high-throughput data, while negative samples can be challenging. An excessive difference between positive and negative samples can lead to model bias, whereas a lack of distinction between them may degrade model performance. And there are almost no directly accessible datasets of IAV-human PPIs, which hinders the further study of influenza A through computational methods. Secondly, effective feature representation to characterize protein sequence information is another important issue in current methods. Many recent studies have demonstrated that a single feature is insufficient to fully characterize sequence information. Therefore, in their study, they utilized a combination of multiple features (Chen et al., 2020; Yu et al., 2020; Gao et al., 2022). However, most of these studies have not detailed how the features were selected or the biological significance of their combination. Finally, the determination of classification algorithms and hyper-parameterization also has a great impact on the overall method.

In this article, firstly, we proposed a new method of negative sample construction to construct a high-quality IAV-human PPI dataset for training prediction model. To better characterize PPI information, protein sequence information was then extracted by integrating the conjoint triad and Moran autocorrelation features. Furthermore, a five-fold cross-validation was employed to compare various machine learning models, the eXtreme Gradient Boosting (XGBoost) model with the best performance was determined to predict the interaction between IAV and human proteins. Finally, based on this XGBoost model, a large number of unknown influenza A virus-human protein interactions were predicted. The GO and pathway analysis showed that these genes were highly associated with influenza A virus infection process, indicating the effectiveness of our method. We hope these predicted IAV-human PPIs and the human target proteins will shed some light in the fields of the research on infection mechanism and anti-virus drug development against IAV. Portions of this text were previously published as part of a preprint (https://www.authorea.com/doi/full/10.22541/au.172115164.41006449/v1).

Materials and Methods

Here, we introduce a computational pipeline (Fig. 1) that is based on a protein sequence-based ML method, allowing us to predict IAV-human PPIs. Firstly, in the data preparation phase, we collected IAV-human PPIs from HPIDB and constructed a high-quality positive dataset through a series of processing, and then applied our proposed degree and dissimilarity-based negative sampling to construct a more realistic and reliable negative dataset. As a result, a high-quality IAV-human PPI dataset was generated, which will be used for validation and training. We also prepared an independent dataset for predicting more IAV-human PPIs. Subsequently, two sequence-based features: conjoint triads and Moran autocorrelation, were used to extract PPI information from protein sequences. Following that, the powerful XGBoost model was used to predict IAV-human PPIs. Finally, a series of systems biology analyses were carried out on the predicted results, which proved that the human proteins in the predicted interactions were involved in the process of influenza A virus infection, indicating the reliability of our method. The overall workflow is shown in Fig. 1.

Figure 1 Schematic framework of our research.

The specific steps are described as follows: Step 1: Data preparation. A training dataset was gathered for model construction and training. An Independent dataset was constructed to predict potential PPI. Step 2: Feature extraction and feature fusion. Using conjoint triad and Moran autocorrelation descriptors to convert the protein sequences into feature vectors and extract PPI information from the sequences. Step 3: Model construction and training. Five-fold cross-validation was used to train the XGBoost model with optimal parameters. Step 4: Prediction and results analysis. The trained model was utilized to predict potential PPIs and the systems biology analysis was performed on the predicted results.

Dataset

Positive dataset

In this study, a total of 10,485 pairs of IAV and human PPIs were acquired by searching the HPIDB3.0 (Host-Pathogen Interaction Database) (Ammari et al., 2016). After applying several filtering criteria, including removing duplicate values, limiting the host to humans, restricting the pathogen to the IAV, setting a minimum protein sequence length of 50 amino acids, and removing non-standard amino acids, a total of 7,011 pairs of IAV and human PPIs were obtained as positive samples for this study. The process is shown in Fig. 2.

Figure 2 Flow chart of positive sample construction.

Negative dataset

Due to the difficulty in obtaining empirical evidence of non-interaction between two proteins, there is no gold standard for constructing negative samples so far. However, the quality of negative samples affects the accuracy of PPI prediction. In response to this problem, many scholars have proposed methods for constructing negative samples. The mainstream methods for constructing negative samples are random sampling, subcellular localization based sampling, and dissimilarity-based sampling. Random sampling is the earliest mainstream method (Zhang et al., 2018). In this method, human proteins and viral proteins from the real PPI are randomly paired together, the original positive combinations are then removed from these pairs, leaving just the remaining pairs as negative samples. This method may take a significant number of positive samples as negative samples. The approach of constructing negative samples based on subcellular localization has been acknowledged and adopted by numerous researchers since its introduction (Jansen & Gerstein, 2004). Subcellular localization data provides insight into the functional position and biological role of proteins within a cell, making it biologically significant for the creation of negative samples. Nevertheless, this approach has its limitations. Ben-Hur & Noble (2006) experimentally demonstrated that although choosing negative examples as pairs of proteins that are localized to different cellular compartments creates high-quality negative examples, it also makes them easier to distinguish from interacting proteins. Dissimilarity-based sampling is proposed by Eid, ElHefnawi & Heath (2015). They hypothesized that viral proteins with high sequence similarity could theoretically interact with a large number of similar host proteins. That is, if there exist human proteins H1, H2 that interact with viral proteins V1, V2 respectively, and if the sequence similarity between V1 and V2 is greater than 80%, then it can be assumed that H1 interacts with V2, and H2 interacts with V2. Conversely, if the sequence similarity between V1 and V2 is less than 20%, then H1-V2, H2-V1 can be considered as non-interacting and can be used as negative samples. This method greatly reduces false negatives and is more biologically meaningful. This method is currently the most recognized method for constructing negative samples and is still used by many researchers today. In addition, Dey, Chakraborty & Mukhopadhyay (2020) proposed a degree-based negative sampling method, which is based on the principle that human proteins with higher degrees are more likely to be attacked by viral proteins. Here, “degree” refers to the number of connections a protein has with other proteins in the human protein-protein interaction network. The degrees of all human proteins are calculated and sorted in ascending order, and proteins with low degrees are selected as negative samples. In Dey, Chakraborty & Mukhopadhyay (2020) the dataset is single human proteins, and this degree-based method of constructing negative samples has not been applied to protein pairs. The prevalent methods for generating negative PPI datasets continue to be subcellular localization-based sampling and similarity-based sampling.

Inspired by these previous works, here we propose a new method, namely, degree and dissimilarity-based negative sampling. The negative dataset obtained by the dissimilarity-based sampling is then further selected according to the degree, and the human proteins involved in it with lower degrees are selected as the final negative samples, thereby significantly enhancing the stringency of the selection process. This approach enhances the overall rigor of the methodology.

The following is how the degree and dissimilarity-based negative sampling is carried out. First, global alignments of pathogen proteins were carried out using the Needleman-Wunsch algorithm on an all-versus-all basis. These alignments employed a linear gap penalty of 10 and utilized the BLOSUM62 scoring matrix (Peris & Marzal, 2011). The obtained comparison scores were then normalized between 0 and 1. The sequence dissimilarity distance between viral proteins V1 and V2 is obtained by subtracting the normalized scores from 1 and the dissimilarity threshold was set at 0.8. For example, for a certain viral protein, we found all other viral proteins that meet the dissimilarity threshold, obtaining all viral proteins with less than 20% similarity to that viral protein. Then the human proteins that originally interact with these viral proteins in the positive sample were selected and randomly paired with the obtained set of viral proteins. After that, we removed the protein pairs that already exist in the positive sample. The remaining pairs were considered as candidate negative samples. Once all candidate negative samples were acquired, the degrees of the human proteins within those samples were calculated. Finally, the protein pairs were sorted in ascending order according to the degree of human proteins involved. An equal number of protein pairs as in the positive samples were selected to determine the negative sample. As a result, a total of 7,011 protein pairs were designated as negative samples. Figure 3 shows the workflow of degree and dissimilarity-based negative sampling.

Figure 3 The workflow of degree and dissimilarity-based negative sampling.

Independent dataset

An independent dataset was prepared to predict potential interactions between human proteins and influenza A virus proteins to get potential human target protein. In this study, we downloaded the full set of IAV and human protein sequences from the UniProt/Swiss-Prot (Boutet et al., 2007) database. After excluding non-standard amino acid sequences and setting a minimum protein sequence length of 50 amino acids, which aligns with the positive sample filtering criteria, we eventually obtained a total of 1,384 IAV proteins and 20,304 human proteins. The IAV and human proteins were randomly combined, as V (1,384) × H (20,304), generating a total of 28,100,736 pairs as full-set samples.

Feature representation

Conjoint triad

Protein-protein interactions are mainly driven by hydrophobic effects, with hydrogen bonding and electrostatic interactions playing a crucial role (Pontremoli et al., 2018). Electrostatic and hydrophobic interactions are affected by the dipole and volume of the amino acid side chain. The conjoint triad is a method proposed by Shen et al. (2007) to classify amino acids based on dipole and side chain volume in order to extract useful information from protein sequences and convert the sequences into feature vectors. It takes into account the properties of an amino acid and its neighbors and regards any three consecutive amino acids as a single unit. Therefore, it is possible to distinguish triads based on the class of amino acids, and by calculating the frequency of each triad type, the PPI information of the protein sequence can be converted into a numerical vector. The process of generating the descriptor vector is described as follows: the 20 amino acids are categorized into seven categories depending on dipole and side chain volumes. The seven categories are {AGV}, {DE}, {FILP}, {KR}, {MSTY}, {HNQW}, and {C}. Considering three consecutive amino acids as one unit, we can get 7 * 7 * 7 = 343 triplet types, then calculate the frequency of each triplet occurrence in the given protein sequence, and finally obtain the 343-dimensional feature vector according to the following formula.

di=fi−min{f1,f2,…,f343}max{f1,f2,…,f343},i=1,2,…,343

Moran autocorrelation

Protein-protein interactions can be categorized into four distinct modes of interaction: electrostatic interactions, hydrophobic interactions, spatial interactions, and hydrogen bond interactions (Arenas et al., 2015). The conjoint triad approach considers the local environment of residues, but it only considers the properties of an amino acid and its two neighboring amino acids. However, information between discontinuous amino acids is equally vital, as these discrete amino acids may be spatially close for protein folding (You et al., 2014; Wang et al., 2017). By considering this information, a more comprehensive understanding of PPIs can be achieved (Guo et al., 2008), so we also use Moran autocorrelation to represent protein sequences. The autocorrelation-based descriptors capture the spatial distribution of local and global properties in proteins by calculating correlations between the physicochemical properties of amino acids at different locations in the protein (Hosseinzadeh et al., 2012). The autocorrelation descriptors also provide access to the physicochemical information contained in the protein sequence, and here eight physicochemical properties of amino acids (Xiao et al., 2015) have been chosen to reflect these interaction patterns as much as possible: normalized average hydrophobicity scales (AccNo. CIDH920105; Cid et al., 1992), average flexibility indices (AccNo. BHAR88010; Bhaskaran & Ponnuswamy, 1988), polarizability parameter (AccNo. CHAM820101; Charton & Charton, 1982), free energy of solution in water (AccNo. CHAM820102), residue accessible surface area in tripeptide (AccNo. CHOC760101; Chothia, 1976), residue volume (AccNo. BIGC670101; Bigelow, 1967), steric parameter (AccNo. CHAM810101; Charton, 1981), and relative mutability (AccNo. DAYM780201; Dayhoff, Schwartz & Orcutt, 1978). The Moran autocorrelation descriptor is defined as follows:

I(d)=1L−d∑i=1L−d⁡(Pi−P¯′)(Pi+d−P¯′)1L∑i=1L⁡(Pi−P¯′)2d=1,2,…,30

where Pi and Pi+d denote the physicochemical properties of the i−th and i+d−th amino acids, and d denotes the interval between two amino acids.

Feature selection

Due to the complexity of protein structures and functions, feature extraction for proteins is more intricate than for DNA and RNA sequences (Tsubaki, Tomii & Sese, 2018). Protein sequence-based features were categorized into five groups: amino acid composition descriptors, such as amino acid composition (AAC) and conjoint triad (CT), Autocorrelation descriptors, such as Geary autocorrelation (Geary) and Moran autocorrelation (Moran); pseudo amino acid composition descriptors, such as pseudo amino acid composition (PseAAC) and amphiphilic pseudo amino acid composition (APseAAC); quasi-sequence-order descriptors, such as quasi-sequence-order (QSO) and sequence-order-coupling number (SOCN); and CTD descriptors, including composition (CTDC), transition (CTDT), and distribution (CTDD) (Liu, 2017). We evaluated features from these categories on the constructed dataset using a wrapper-based feature selection method, identifying the best-performing features within each group. Subsequently, feature combination and ablation experiments were conducted to determine the optimal feature set.

Optimization and training of the XGBoost model

XGBoost (Chen & Guestrin, 2016) is an integrated learning algorithm based on gradient boosting trees, whose core idea is to build a powerful prediction model by combining multiple decision trees. The prediction accuracy of the model is improved by iteratively training multiple decision trees. In each iteration, XGBoost calculates the gradient and second-order derivative of the loss function of the current model, and then constructs a new decision tree to fit the current negative gradient residuals. The step-by-step approach employed by XGBoost enables it to effectively adapt to complex binary classification problems, making it highly suitable for such scenarios (Wu, Li & Ma, 2021). The objective function of XGBoost can be expressed as:

Obj(t)=∑i=1n⁡l(yi,y^i(t))+∑i=1t⁡Ω(fi)

where yi is the true value, y^i is the predicted value, ∑i=1n⁡l(yi,y^i(t)) is the loss function and Ω(fi) is the canonical term.

To optimize the model’s performance and enhance its generalization ability, an iterative process was employed to test different parameter combinations. Specifically, the learning_rate parameter was tested within the range of 0.01 to 0.2 in steps of 0.01, the max_depth parameter within the range of 3 to 10 in steps of 1, and the n_estimators parameter within the range of 50 to 500 in steps of 10. For the optimization, 80% of the dataset obtained from the previous steps was used for training while the remaining 20% was for model validation. By employing a five-fold cross-validation approach with grid search, the final optimal parameters determined were: learning_rate = 0.1, max_depth = 7, and n_estimators = 310. These parameter values have been identified as the most suitable for achieving optimal performance in the classification task.

To evaluate the performance of XGBoost, we compared it with other machine learning models. Some widely-used PPI classifiers such as Random Forest (RF) (Chen & Liu, 2005; Wang et al., 2018), support vector machine (SVM) (Chatterjee et al., 2011; Cui, Fang & Han, 2012; You et al., 2015), logistic regression (LR) (Qi, Bar-Joseph & Klein-Seetharaman, 2006), K nearest neighbors (KNN) (Guarracino & Nebbia, 2010) and other tree models such as extremely randomized trees (ET) (Yu et al., 2021; Peng et al., 2018) and adaptive boosting (Adaboost) (Mei & Zhu, 2014) were compared. These model parameters and optimization are detailed in the Supplemental File.

Specifically, the training dataset in our study maintained a balanced ratio of positive and negative samples at 1:1. We extracted protein sequence information using two features, namely CT+Moran. The XGBoost model was trained using a five-fold cross-validation technique, with optimal parameters configured for the task. Subsequently, the final trained model was saved.

Prediction of IAV-human protein-protein interactions

In this study, we not only evaluated the performance of our proposed method through various tests but also applied it to prediction in practice. We prepared an independent dataset to predict more PPIs. The constructed independent dataset encompasses all IAV proteins and human proteins. It consists of a total of 28,100,736 protein pairs obtained by randomly pairing IAV proteins and human proteins together. Similarly, these 28,100,736 protein pairs were characterized using CT+Moran features. The model trained in the previous section was used to make predictions on this independent dataset. The final predicted results of the model were obtained in the form of the corresponding protein pair ID along with its probability of being a positive PPI. Due to the vast amount of data in the prediction, it is necessary to establish probability thresholds to determine potential PPIs based on the overall results, ensuring rigorous analysis. For instance, in the study conducted by Dey, Chakraborty & Mukhopadhyay (2020), the threshold setting was carefully considered while predicting potential human protein targets of SARS-CoV-2 and finally set the threshold to 0.7 in order to get a reasonable number of predictions. In order to obtain potential human protein targets of the IAV with high confidence, we further screened the human proteins involved in the predicted PPIs and selected human proteins with a degree greater than 0.5*virus number (human proteins targeted by more than half of the IAV), the corresponding high-confidence target genes were obtained, and subsequently verified using gene ontology and KEGG pathway enrichment analysis, as well as network topology analysis.

Performance measures

In order to construct more effective models with more significant features, a five-fold cross-validation approach was utilized for training. Five-fold cross-validation involves randomly dividing the samples into five subsets. Four of these subsets are used as the training set, while the remaining subset serves as the test set. This process is repeated five times, with each time employing a different subset as the test set. Then the fully trained model is validated on an independent dataset to obtain more robust results. Various measurements, including accuracy, precision, specificity, F1-score, recall, and Matthew’s correlation coefficient (MCC), were calculated using balanced datasets with equal numbers of positive and negative samples. These measures are formulated as follows:

Accuracy=TP+TNTP+TN+FP+FN

Precision=TPTP+FP

Specifity=TNFP+TN

F1_score=2∗(Recall∗Precision)(Recall+Precision)

Recall=TPTP+FN

MCC=(TP∗TN−FP∗FN)(TP+FP)∗(TP∗FN)∗(TN+FP)∗(TN+FN)

where TP, FP, TN, and FN are true positive, false positive, true negative, and false negative, respectively.

Systems biology analysis

Gene ontology and pathway enrichment analysis

The functional enrichment analysis of genes by gene ontology (GO) and pathway analysis were performed on target genes. The GO and KEGG (Kyoto Encyclopedia of Genes and Genomes) pathway enrichment analysis was performed via the DAVID (Database for Annotation, Visualization and Integrated Discovery) online website (Sherman et al., 2022), and the top 20 items were selected for visualization via the ggplot2 package in R (Ito & Murphy, 2013). Significant GO and KEGG pathways were selected using a statistical threshold criterion of adjusted P-value < 0.05.

Identification of hub genes and PPI network analysis

The target genes obtained through our model prediction were mapped on the STRING database (Szklarczyk et al., 2022), and the PPI was constructed by selecting a threshold of confidence >0.90. Network analysis was performed using Cytoscape (Smoot et al., 2010) to filter out the genes with degree ≥8, which were identified as potential hub genes (Tian et al., 2019). To better identify hub genes, the top 20 hub genes were screened from this network based on connectivity by CytoHubba plugin for Cytoscape. These high confidence hub genes are considered as high confidence biomarkers of the human pathway of Influenza A virus infection.

Results and discussion

Performance of different negative sampling methods

In this study, we proposed a novel and rigorous negative sampling method that produces a more reliable negative interaction dataset compared to previous approaches. To validate the effectiveness of our method, we compared the datasets generated by our proposed method with those obtained from alternative negative sampling techniques using various models. The alternative methods considered include random sampling, dissimilarity-based sampling, degree-based sampling, and subcellular localization-based sampling. To prevent statistical discrepancies, we employed a balanced dataset with a 1:1 ratio of positive to negative samples. All datasets were divided into two parts, with 80% being extensively trained using five-fold cross-validation. After training, validation was conducted on the remaining 20% independent dataset, and the performance on this independent dataset was compared. The comparison results are presented based on accuracy rates, and the corresponding outcomes are displayed in Table 1. It demonstrates that our negative sampling approach not only improves prediction performance but also exhibits biological relevance. Specifically, this method is based on two key principles: firstly, viral proteins with high sequence similarity are less likely to interact with the same host proteins, and secondly, host proteins with lower degrees of connectivity have a lower probability of interacting with viral proteins. Compared to other individual methods, our approach avoids the bias commonly observed with subcellular localization methods, while also outperforming several widely used approaches in terms of performance.

Table 1 Comparison of accuracy (%) of different machine learning algorithms on 1:1 positive: negative training dataset considering random sampling, dissimilarity-based sampling, degree distribution and degree and dissimilarity-based of preparing negative.

	Random sampling	Dissimilarity-based sampling	Degree-based sampling	Degree and dissimilarity-based	
XGBoost	80.80%	75.83%	95.72%	96.84%	
Random forest	81.21%	76.11%	92.58%	94.63%	
LightGBM	81.45%	76.18%	93.66%	95.29%	
ExtraTrees	81.50%	76.29%	91.50%	93.74%	
AdaBoost	81.09%	75.46%	89.89%	92.11%	
SVM	81.11%	75.45%	89.83%	91.92%	
LR	80.98%	75.02%	89.21%	91.15%	

Performance of feature representation

We compared the features from five different categories on our constructed dataset. The results are illustrated in the Fig. 4. Among all the categories, it is evident that amino acid composition and autocorrelation provide the most accurate description of our dataset. The conjoint triad feature within the amino acid composition category demonstrates superior performance, while Moran autocorrelation stands out as the optimal feature within the autocorrelation category. Previous studies have consistently shown that the fusion of multiple features offers a more comprehensive understanding of the information associated with protein interactions (Chen et al., 2020; Yu et al., 2020; Chen et al., 2019; Gao et al., 2022). The best features in each category were compared with combined CT+Moran features and the results are shown in the Fig. 5. It can be seen that CT+Moran outperforms the best single feature in all categories.

Figure 4 ACC, MCC, and F1-score of different features from different categories on independent dataset.

Figure 5 The comparison of accuracy, recall, F1-score and MCC of best-performing features in each category with CT+Moran.

In this study we used two types of features to convert protein sequences into numerical vectors, the conjoint triad and Moran autocorrelation, respectively. By integrating these two features, we obtained a more comprehensive representation of protein interaction information, as demonstrated in previous studies (Chen et al., 2020; Yu et al., 2020; Chen et al., 2019; Gao et al., 2022). To further validate the efficacy of combining these features, we conducted an ablation experiment specifically targeting this feature fusion. The results of this experiment are presented in the accompanying Table 2. The results show that the accuracy when fusing the two features is 91.37%, with a loss of 2.75% accuracy when CT is removed and 1.07% accuracy when Moran is removed. In terms of precision, removing the CT feature results in a reduction of 0.78%, while excluding the Moran feature leads to a decrease of 0.3%. For recall, the removal of the CT feature causes a reduction of 4.49%, whereas excluding the Moran feature results in a decrease of 1.4%. Regarding the F1-score, removing the CT feature leads to a reduction of 2.72%, and excluding the Moran feature results in a decrease of 0.87%. For MCC, the removal of the CT feature causes a reduction of 0.0533, while excluding the Moran feature results in a decrease of 0.021. This suggests that the feature combination we selected effectively represents both the continuous and discontinuous information of amino acids, and demonstrates potential in the prediction task of influenza virus and human host protein interactions among all the features we tested.

Table 2 Results of ablation study of CT+Moran fusion feature.

	CT+Moran	CT	Moran	
Accuracy (%)	91.37%	90.30%	88.62%	
Precision (%)	92.15%	91.85%	91.37%	
Recall (%)	90.15%	88.75%	85.66%	
F1_score (%)	91.14%	90.27%	88.42%	
MCC	0.8275	0.8065	0.7742	

Determination of the ML model

To build an effective PPI prediction model, the determination of the classifier is crucial. In our study, we concatenated the protein pair vectors encoded by CT and Moran, resulting in a 1,166-dimensional representation of PPIs, and eXtreme Gradient Boosting (XGBoost) was determined as the final classifier. To evaluate the performance of XGBoost, we compared it with other machine learning models on the 1,166-dimensional dataset. Some widely-used PPI classifiers such as RF (Chen & Liu, 2005; Wang et al., 2018), SVM (Chatterjee et al., 2011; Cui, Fang & Han, 2012; You et al., 2015), LR (Qi, Bar-Joseph & Klein-Seetharaman, 2006), KNN (Guarracino & Nebbia, 2010) and other tree models such as ET (Yu et al., 2021; Peng et al., 2018) and Adaboost (Mei & Zhu, 2014) were compared. The radial basis kernel function was used in SVM. L2 regularization with a regularization factor of 6 was used in LR. The number of neighbors in k-NN was configured as 2. The n_estimators of AdaBoost, RF and XGBoost were all configured as 310. Figure 6 shows the accuracy of different models for each fold in the five-fold cross-validation. It is evident in Fig. 6 that both SVM and XGBoost exhibit the superior performance. Meanwhile, XGBoost slightly outperforming SVM. To further highlight the superior performance of XGBoost, we compared the average accuracy, precision, recall, F1-score, and MCC across different models in the five-fold cross-validation. The results are presented in Fig. 7, clearly demonstrating that XGBoost outperforms SVM in all aspects, making it the top-performing model among all the evaluated models.

Figure 6 Accuracy of each fold in the five-fold cross-validation of different models.

Figure 7 Accuracy, Precision, Recall, F1-score and MCC for different models on independent dataset.

Comparison with other methods

Applying our method to other datasets

To provide a more objective evaluation of the predictive performance of our constructed model, we conducted a series of comparative experiments. Firstly, we obtained the datasets used by existing PPI prediction methods proposed in the literature. Subsequently, we applied our method to these datasets, utilizing the CT+Moran features for protein sequence extraction and employing the XGBoost model for prediction. The results obtained using our method were then compared with those reported in the original article.

The two datasets published by Zhou et al. (2018) have been widely adopted as benchmarks for evaluating the performance of state-of-the-art models in viral-human PPI prediction tasks. We refer to these datasets as Zhou’s H1N1 and Zhou’s Ebola, with each dataset named after the virus represented in the respective test set.

In Zhou’s H1N1 dataset, the training set contains 10,955 true PPIs between humans and any virus other than H1N1, along with an equal number (10,955) of negative interaction samples. The test set consists of 381 real PPIs between humans and the H1N1 virus and 381 negative interactions. It can be seen in Table 3 that the sensitivity, specificity, accuracy, and MCC of our method on Zhou’s H1N1 are 91.60%, 70.07%, 80.83%, and 0.631. Their original method’s performance on sensitivity, specificity, accuracy, and MCC are 66.39%, 65.98%, 66.19%, and 0.324. The results presented in Table 3 demonstrate the superior performance of our method, particularly in terms of sensitivity and MCC, when compared to the original research approach.

Table 3 Comparison of our method and Zhou’s method on Zhou’s H1N1 dataset.

	Sensitivity (%)	Specificity (%)	Accuracy (%)	F1_score (%)	MCC	
Zhou’s	66.39	65.98	66.19	–	0.324	
Our	91.60	70.07	80.83	82.70	0.631	

In Zhou’s Ebola dataset, the training set contains 11,341 true PPIs between humans and any virus other than Ebola and an equal number (11,341) of negative interaction samples. The test set contains 150 true PPIs between humans and Ebola viruses and 150 negative interactions. Table 4 displays the performance metrics of our method on Zhou’s Ebola dataset, including sensitivity, specificity, accuracy, F1-score, and MCC. Our method achieves a sensitivity of 96.66%, specificity of 66.00%, accuracy of 81.33%, F1-score of 83.81%, and MCC of 0.658. In comparison, the original method outlined by Zhou et al. (2018) reports a sensitivity of 90.67%, specificity of 65.33%, accuracy of 78.00%, and MCC of 0.579. The results indicate that our method is better than the original method in all aspects.

Table 4 Comparison of our method and Zhou’s method on Zhou’s Ebola dataset.

	Sensitivity (%)	Specificity (%)	Accuracy (%)	F1_score (%)	MCC	
Zhou’s	90.67	65.33	78.00	–	0.579	
Our	96.66	66.00	81.33	83.81	0.658	

We also assessed the performance of our method on the dataset developed by Prasasty et al. (2021), referred to as Prasasty’s bacteria. Prasasty’ bacteria contain three human pathogens, Bacillus anthracis, Yersinia pestis, and Francisella tularensis. They utilized two of the bacteria as the training set, while the remaining one was used as the test set for their analysis. In our evaluation, we adopted Bacillus anthracis and Yersinia pestis as training sets, Francisella tularensis as the test set. Therefore, the training set contains 6,354 positive interactions between humans and bacteria, and the corresponding negative interactions. The test set contains 1,187 positive interactions between humans and Francisella tularensis, as well as 1,187 negative interactions. The outcomes are shown in Table 5. The method utilized in the original article by Prasasty et al. (2021) achieved the following performance on this dataset: sensitivity of 74.56%, specificity of 97.83%, accuracy of 95.84%, and precision of 76.36%. However, when applied to our dataset, our method achieves a sensitivity of 93.11%, specificity of 95.28%, accuracy of 93.25%, and precision of 95.07%. Although our model exhibits slightly lower specificity and accuracy, it displays higher sensitivity and precision. As a result, our method is superior overall.

Table 5 Comparison of our method and Prasasty’s method on Prasasty’s bacteria dataset.

	Sensitivity (%)	Specificity (%)	Accuracy (%)	F1_score (%)	MCC	Precision (%)	
Prasasty’s	74.56	97.83	95.84	–	–	76.36	
Our	93.11	95.28	93.25	93.38	0.870	95.07	

Additionally, it is worth mentioning that the three datasets we evaluated were not based on a single virus-human PPI dataset. In these datasets, the training and test sets involve different viruses, leading to low sequence similarity between their proteins. In contrast, our dataset consists entirely of influenza A virus-human PPI data, which might suggest potential dataset bias. However, our method demonstrated strong generalization ability, consistently outperforming others across diverse datasets. This robustness highlights its potential as a foundational model for PPI prediction across viruses.

Comparison with other methods on our dataset

In order to evaluate our method more comprehensively, we also applied the proposed approach by others to our dataset and conducted a comparative analysis with our own method’s results. Specifically, we employed the method presented in Denovo (Eid, ElHefnawi & Heath, 2015), which utilized CT for protein feature extraction and SVM for prediction. The comparative outcomes are summarized in Table 6. Our original results exhibit superior performance compared to Denovo. In particular, our results demonstrate an accuracy of 96.89%, precision of 98.79%, recall of 94.85%, F1-score of 96.78%, and MCC of 0.9386. These values are 6.41%, 7.29%, 6.39%, 6.83%, and 0.1291 higher than Denovo in terms of accuracy, precision, recall, F1-score, and MCC, respectively.

Table 6 Comparison of our method and Dovono on our training dataset.

	Accuracy (%)	Precision (%)	Recall (%)	F1_score (%)	MCC	
Denovo’s	90.48	91.50	88.46	89.95	0.8095	
Our	96.89	98.79	94.85	96.78	0.9386	

Gene ontology and pathway enrichment analysis

In this study, we established a threshold value of 0.95 to determine the predicted results of the model. To better assess the reliability of the predicted influenza virus-specific human protein targets, we also stipulated that human proteins must be targeted by over half of the total number of viruses. This threshold of 0.5 is commonly used in statistical analysis. As a result, we identified a total of 32,855 interactions, from which we identified 3,269 potential human target proteins, and yielded 2,995 target genes with high confidence. The predicted result is shown in the Supplemental Files. To further understand and validate the high-confidence target genes obtained from the model, GO and KEGG pathway analysis was performed in this study. The corresponding results can be observed in Fig. 8. The biological process is significantly enriched mainly in replication, transcription and translation (A). These biological processes are highly associated with viral processes. Viral processes are a number of biological processes in which viruses are involved, including infection of host cells, replication of the viral genome and assembly of daughter viral particles (Huang et al., 2019). IAV enter host cells mainly through transmembrane transport, such as cytokinesis and vesicular transport. The influenza virus genome undergoes transcription and replication in the host cell nucleus, followed by exit and transit, and final assembly and release (Nuwarda, Alharbi & Kayser, 2021). The cellular component is mainly enriched in the nucleus, cytoplasm and extracellular vesicles (B), which are highly similar to those involved in the life cycle of the influenza A infected host. The molecular function is mainly enriched in binding-related functions, especially protein-related binding (C), which further validates the plausibility of these high confidence target genes.

Figure 8 (A–D) Results of GO and KEGG pathway analysis.

The KEGG pathway is significantly enriched in infection-related pathways (viral carcinogenesis, prion diseases, coronavirus disease-COVID-19), neurological disease-related pathways (amyotrophic lateral sclerosis, Alzheimer disease, Huntington disease and Parkinson disease), immune-related pathways (systemic lupus erythematosus, neutrophil extracellular trap formation and necroptosis), alcoholism and signaling-related pathways (D). In terms of viral infection-related pathways, IAV and SARS-CoV-2 share similar infection pathways and associated pathogenesis (Ilyicheva & Gureyev, 2021). Meanwhile, recent studies have suggested a possible association between IAV and a variety of neurological disorders (Levine et al., 2023), particularly between influenza and Parkinson’s disease (Hoffman & Vilensky, 2017). In addition to this, IAV infection is often accompanied by an immune and inflammatory response (Wan et al., 2022), in which IAV can cause apoptosis of epithelial cells in the upper respiratory tract, triggering immunity in the respiratory mucosa and causing an inflammatory response in vivo (Zhu et al., 2022). The above GO and pathway analyses suggest that these genes are highly associated with IAV infection.

PPI network analysis with identified hub genes

By uploading the obtained target genes to the STRING database, 427 potential hub genes (PPIhub) were finally obtained based on degree ≥8. To better identify the hub genes obtained from the model, a PPI network of the top 20 highly connected genes was constructed based on the size of the degree using Cytoscape’s CytoHubba plugin. The importance ranking of these 20 pivotal genes is given based on the network interactions between these genes, based on degree (Table 7). All genes except UBA52 appeared in the positive sample dataset. However, previous studies have shown that knockdown of UBA52 in chicken cells resulted in reduced viral titers in the offspring, confirming the important function of UBA52 in H5N1 influenza A virus infection (Ghobadi et al., 2019). The other 19 genes were all genes encoding ribosomal proteins. RPS11 and RPS8 have been shown to affect influenza A infection (Murray, Sheng & Rubin, 2014) and RPS3 is thought to be a gene involved in viral transcription and translation (Cui et al., 2020). Most of these proteins are thought to be ribosomal proteins involved in influenza virus RNA transcription and viral mRNA translation, and also have high connectivity in the human protein interactions network (Hegde et al., 2012). This suggests that our model may be biased to focus on learning sequence features of human host proteins that affect transcription and translation of the A virus, a point that may be related to the dataset we constructed, particularly the original positive sample dataset. Furthermore, it is also possible that the degree of filtering we performed during the negative sample construction resulted in the majority of our positive predictions also being biased towards the core proteins of the human protein interactions network.

Table 7 Ranking of the importance of the 20 pivotal genes.

Rank	Gene symbol	Uniprot	Degree	
1	RPS27A	P62979	140	
2	RPS11	P62280	134	
3	RPS5	P46782	128	
4	UBA52	P62987	126	
5	RPS18	P62269	124	
6	RPS6	P62753	122	
7	RPS8	P62241	121	
8	RPS23	P62266	120	
8	RPS9	P46781	120	
8	RPS7	P62081	120	
8	RPS16	P62249	120	
8	RPS3A	P61247	120	
13	RPS14	P62263	119	
13	RPS15A	P62244	119	
15	RPS13	P62277	118	
15	RPS24	P62847	118	
17	RPS28	P62857	117	
18	RPS2	P15880	115	
18	RPS27	P42677	115	
18	RPS3	P23396	115	

Conclusion

In our study, we constructed a high-quality IAV-human PPI dataset and employed XGBoost to predict interactions between influenza A virus proteins and human proteins. By leveraging CT and Moran autocorrelation features, our method effectively captured PPI information from protein sequences. Extensive experimental comparisons demonstrated that our approach outperforms other methods for this task, and its strong performance across diverse datasets further underscores its potential as a foundational tool for predicting PPIs involving various viruses. In a five-fold cross-validation of our benchmark dataset, the model achieved an accuracy of 96.89%, precision of 98.79%, recall of 94.85%, F1-score of 96.78%, and MCC of 0.9386.Our model ultimately predicts 32,855 PPIs, involving 3,269 potential target proteins corresponding to 2,995 target genes. The GO and pathway analysis showed that these genes were highly associated with influenza A virus infection. In the network topology analysis, the predicted proteins exhibited high connectivity within the human protein interactions network. This finding further reinforces the credibility and reliability of our prediction results.

While this study achieved promising prediction results, we acknowledge that the current approach primarily relies on sequence-based feature representations, which are inherently limited in capturing the full complexity of protein information. To address this, future work will focus on integrating structural data, multi-omics approaches, and phenotypic information to develop more robust and biologically meaningful predictive models. Ultimately, we hope that this research will help biologists recognize possible associations between influenza A virus and human proteins, and facilitate the development of antiviral drugs.

Supplemental Information

Supplemental Information 1 Appendix A-Parameter setting and optimization of different models.

Supplemental Information 2 Appendix_B-Predicted_ppi.

Supplemental Information 3 Appendix_C-Predicted_human_uniprotid.

Supplemental Information 4 Appendix_D-Potential_hub_genes.

Additional Information and Declarations

Competing Interests

The authors declare that they have no competing interests.

Author Contributions

Binghua Li conceived and designed the experiments, performed the experiments, authored or reviewed drafts of the article, and approved the final draft.

Xin Li conceived and designed the experiments, performed the experiments, authored or reviewed drafts of the article, and approved the final draft.

Xiaoyu Li analyzed the data, authored or reviewed drafts of the article, and approved the final draft.

Li Wang analyzed the data, prepared figures and/or tables, and approved the final draft.

Jun Lu analyzed the data, prepared figures and/or tables, and approved the final draft.

Jia Wang conceived and designed the experiments, authored or reviewed drafts of the article, and approved the final draft.

Data Availability

The following information was supplied regarding data availability:

The source codes and datasets are available at GitHub and Zenodo:

- https://github.com/HVPPIlab/IVA-Human-PPI/.

- li,. binghua. (2024). IVA-Human-PPI. Zenodo. https://doi.org/10.5281/zenodo.14273568.

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
