# Peer review of "Prediction of influenza A virus-human protein-protein interactions using XGBoost with continuous and discontinuous amino acids information"

_PeerJ, doi:10.7717/peerj.18863_

## Round 0.1 · original submission · Major Revisions

I am sure that the authors are able to make all necessary changes. The issues related to hyper-parameterization must be discussed.

Reviewer 1 ·

Basic reporting

This manuscript focuses on predicting Influenza A Virus (IAV)-human protein-protein interactions (PPIs) using machine learning methods based on protein sequence information. It demonstrates several notable strengths, including a well-structured organization, clearly articulated research background and significance, professionally presented figures and data visualization. However, there are several areas requiring improvement. The methodology section needs more detailed description, particularly regarding the negative sample construction process. Additionally, some technical terms, such as "fusion features," require clearer explanation for better reader comprehension. The manuscript also lacks statistical significance analysis for several experimental results, and the accessibility of raw data needs to be better addressed. These improvements would enhance the overall clarity and scientific rigor of the paper.

Experimental design

1.Biological rationale for negative sample construction needs stronger justification.
2.Feature selection process lacks detailed explanation and theoretical support.
3.Model parameter selection criteria need more justification.
4.Experimental reproducibility validation should be supplemented.

Validity of the findings

1.More experimental validation needed for computational predictions.
2.Deeper biological interpretation of prediction results required.
3.Potential model bias towards transcription and translation-related proteins.

Additional comments

1.Add experimental validation of predictions.
2.Develop user-friendly prediction tool interface.
3.Provide detailed usage documentation.

Reviewer 2 ·

Basic reporting

Authors developed a sequence-based machine learning method for predicting PPI and trained/tested on a custom dataset. The method’s performance in terms of accuracy (96.89%), precision(98.79%), recall(94.85%) and F1-score of (96.78%) is impressive.

The paper is nicely organized; motivation is captured well, relevant literature adequately shared, methods are clearly stated, and results overall is capturing different aspects.

Experimental design

My main concerns are around 1) hyperparameterization experiments and their results that led to selection of XGBoost 2) the independent dataset generation not considering subcellular localization.

The detailed list of my concerns suggestions with line numbers/context is provided under "Additional comments".

Validity of the findings

The findings and conclusions are clear and supported by results. Underlying data is provided and replication is possible.

Additional comments

My concerns/suggestions with line numbers/context include:

Line 54: There are many traditional methods - > an alternative word replacing traditional maybe experimental and revising the sentence is recommended.

Line 83-87: In a paragraph discussing SOA methods, one method’s results around dengue virus is described exclusively. I think either this level of detail is mentioned for other methods or removed here for consistency.

Line 94: Highlights “effective”-ness of feature representation, yet it is not clear what makes a representation effective or not is not described or understood. Additional explanation would improve the manuscript.

Line 99: “The determination of classifier” probably is referring to selection of classification algorithms. This can be clarified. But on top of algorithm selection, hyperparameterization of these algorithms is also important.

Line 160: The cited ”degree-based” approach deserves further explanation. What is meant by degree? Given “degree” is also employed in proposed approach, more details can be provided.

Line 189(Independent dataset): As per description provided, the independent set creation does not consider subcellular localization. Could this be introducing interactions that cannot be observed in vivo and inflating the negative set? A discussion would be nice.

Line 193: What is meant by “non-standard amino acid sequences”? Is decided based on length or content? Some clarification would be nice to have.

Line 239(Optimization and Training of the XGboost model): Other machine learning algorithms are not discussed as part of methods. Perhaps some content in section 3.3 could move to section 2.3.

Line 360(Determination of the ML model) The hyperparameters of algorithms don’t seem to be experimented or experiments are documented. It is as if each algorithm is tried with one parameter. Given SVM is presenting a comparable performance, it is worthwhile to see all hyperparameterization performance results i.e. in an appendix.

Line 379: Section title can be “Applying our method to other datasets”

Line 396: table 3 -> Table 3 (upper case)

Line 401-406: What could be the explanation of low specificity of the model on Zhou’s set when compared to other datasets?

Line 407: we -> We (upper case)

Line 421: Section title can be Comparison with other methods on our dataset

Line 430: The gene ontology and pathway enrichment analysis identified human nuclear proteins’ physical interaction with viral proteins. Is this biologically plausible for all occasions? Does this call for a subcellular localization-based sampling (see above comment on Line 189)

Line 484-487: The sentence is too long. It would be nice to rephrase.

Reviewer 3 ·

Basic reporting

Clarity and Typos:
Typo in line 85: "These findings."
Could motivate the problem statment in lines 90-92, particularly by elaborating on how the construction of positive and negative samples has been a limitation in prior studies.
Clarify the use of “fusion features” in line 97, as it’s unclear if it refers to multiple features being combined.
Consider rephrasing lines 97-99 for clarity, particularly the phrase “what the significance of combination is.”

Background and Citations:
Cite previous studies again in line 349 to support statements.

Experimental design

The study addresses a key challenge in predicting interactions between viral and cellular proteins by tackling two major bottlenecks: defining negative sample sets, which is crucial for training classifiers, and extracting features that can be used in models. To improve negative sample definition, the authors combined two existing methods—degree-based and dissimilarity-based negative sample preparation—and demonstrated that this combined approach enhances the accuracy of machine learning algorithms.

For feature extraction, the authors utilized conjoint triad and Moran autocorrelation methods to represent protein sequences and showed that combining these features yields better results than using each individually. By integrating the combined negative sample approach with the fusion of features from conjoint triad and Moran methods, the authors tested various classifiers, finding that XGBoost performed best for accurately predicting PPI. This new pipeline was also compared to previously published benchmark datasets, and results indicate that it outperforms prior methods. Overall, this is a well-executed study that addresses a few key challenges in predicting viral and cellular protein interactions. However, there is room for further exploration to discover novel biological insights using this pipeline.

Validity of the findings

In Section 3.1, provide insights and conclusions drawn from Table 1 to help interpret the data effectively. This will make the results clearer and more meaningful to readers.

In line 433, clarify why the assumption was made that human proteins must be targeted by over half of the total number of viruses. Justifying this assumption will strengthen the study's rationale.

Of all the protein interactions identified (“we identified a total of 32,855 interactions, from which we identified 3,269 potential human target proteins, yielding 2,995 target genes with high confidence”), discuss how many of these interactions are novel compared to those already present in the training dataset.

If the GO/KEGG analysis and hub protein identification are based solely on novel interactions, can any new pathways be uncovered that could provide insights into previously unexplored biological mechanisms?

The study would be further strengthened by exploring feature importance within the model to pinpoint which features most significantly contribute to predictive performance.

Conclusion Improvement:

Enhance the conclusion by discussing the limitations of the current methodology and suggesting potential future steps to improve PPI prediction accuracy.

Discuss how experimental validation can support the development of better predictive models, potentially leading to more reliable and biologically relevant predictions.

Could also discuss the potential of foundational models for PPI across viruses.

---

## Round 0.2 · accepted · Accept

I have been glad to see that you addressed all comments and suggestions of all reviewers.

Reviewer 2 ·

Basic reporting

No comment

Experimental design

No comment

Validity of the findings

No comment

Additional comments

Authors responded to all of the comments from my previous review adequately. In particular, clarifications on hyperparameterization and the role of subcellular localizations in negative set construction are helpful. Hence, I have no further comments on the manuscript.